# Anti-Inflammatory Effects of (9Z,11E)-13-Oxooctadeca-9,11-dienoic Acid (13-KODE) Derived from *Salicornia herbacea* L. on Lipopolysaccharide-Stimulated Murine Macrophage via NF-kB and MAPK Inhibition and Nrf2/HO-1 Signaling Activation

**DOI:** 10.3390/antiox11020180

**Published:** 2022-01-18

**Authors:** Yu-Chan Ko, Hack Sun Choi, Su-Lim Kim, Bong-Sik Yun, Dong-Sun Lee

**Affiliations:** 1Interdisciplinary Graduate Program in Advanced Convergence Technology & Science, Jeju National University, Jeju 63243, Korea; uchan@jejunu.ac.kr (Y.-C.K.); choix074@jejunu.ac.kr (H.S.C.); ksl1101@jejunu.ac.kr (S.-L.K.); 2Subtropical/Tropical Organism Gene Bank, Jeju National University, Jeju 63243, Korea; 3Bio-Health Materials Core-Facility Center, Jeju National University, Jeju 63243, Korea; 4Practical Translational Research Center, Jeju National University, Jeju 63243, Korea; 5Faculty of Division of Biotechnology, College of Environmental and Bioresource Sciences, Jeonbuk National University, Iksan 54596, Korea; bsyun@jbnu.ac.kr; 6Faculty of Biotechnology, College of Applied Life Sciences, Jeju National University, Jeju 63243, Korea

**Keywords:** (9Z,11E)-13-Oxooctadeca-9,11-dienoic acid (13-KODE), *Salicornia herbacea* L., inflammation, Nrf-2 (Nfe2I2), macrophage, antioxidant

## Abstract

Glasswort (*Salicornia herbacea* L.) is a halophyte that exhibits antioxidant and antidiabetic effects. Only a few studies have been conducted on its antioxidant effects. Here, we isolated an antioxidant using an activity-based purification method, and the resulting compound was identified as (9Z,11E)-13-Oxooctadeca-9,11-dienoic acid (13-KODE). We investigated its ability to suppress inflammatory responses and the molecular mechanisms underlying these abilities using lipopolysaccharide-stimulated RAW 264.7 macrophage cells. We studied the anti-inflammatory effects of 13-KODE derived from *S. herbacea* L on RAW 264.7 macrophages. 13-KODE inhibited lipopolysaccharide (LPS)-induced nitric oxide (NO) production by suppressing inducible NO synthase and suppressed LPS-induced tumor necrosis factor and interleukin-1β expression in RAW 264.7 macrophages. LPS-mediated nuclear localization of NF-κB and mitogen-activated protein kinase activation were inhibited by 13-KODE. 13-KODE significantly reduced LPS-induced production of reactive oxygen species and increased the expression of nuclear factor erythroid-2 like 2 (Nfe2I2) and heme oxygenase 1. Overall, our results indicate that 13-KODE may have potential for treating inflammation.

## 1. Introduction

Inflammation is the complex biological response of body tissues against pathogens and damaged cells [1]. It is a protective response involving immune cells and molecular mediators. The function of inflammation is to eliminate necrotic cells and damaged tissues. Acute inflammation is considered a part of innate immunity and represents the first line of defense against foreign bacteria and dangerous molecules [2]. Infectious agents and cell damage activate inflammatory cells and induce inflammatory signaling pathways, such as NF-κB, mitogen-activated protein kinase (MAPK), and JAK-STAT signaling pathways [3]. Lipopolysaccharide (LPS), an endotoxin derived from the outer membrane of *Escherichia coli,* induces inflammation and is used to develop disease models to examine the anti-inflammatory effects of drugs and natural compounds [4,5]. Macrophages exposed to LPS produce proinflammatory mediators, cytokines, and reactive oxygen species (ROS) [6,7]. The major proinflammatory mediators consist of nitric oxide (NO) and prostaglandin E2, which are produced by inducible NO synthase (iNOS) and cyclooxygenase-2, respectively [8,9]. LPS-treated macrophages induce tumor necrosis factor (TNF)-α and interleukin (IL)-1β, and these molecules contribute to various inflammatory diseases [10,11]. Furthermore, LPS-stimulated macrophages secrete proinflammatory cytokines and growth factors [12]. LPS exposure results in the production of proinflammatory mediators and proinflammatory cytokines through NF-κB activation [13,14]. Inflammation-related gene expression of immune cells is regulated by the NF-κB pathway [13,15]. LPS stimulation promotes the nuclear translocation of NF-κB p65 through IκB-α reduction. MAPKs regulate cell proliferation, cycle arrest, migration, differentiation, senescence and apoptosis [16]. Inflammation induces ROS production and decreases the production of antioxidant enzymes [17,18]. The crosstalk between inflammation and oxidative stress is important in diseases. The evidence of this crosstalk and the protective effects of natural compounds against oxidative stress and inflammatory response has been shown in previous studies [19,20,21,22,23,24,25]. Heme oxygenase-1 (HO-1) is regulated by nuclear factor erythroid-2 like 2 (Nfe2I2) and cleaves heme to form biliverdin, which is subsequently converted into bilirubin by biliverdin reductase. HO-1 is associated with antioxidant, anti-inflammatory, and cytoprotective functions and has emerged as a target molecule with therapeutic implications [26]. Nfe2I2 plays a central role against inflammation and oxidative damage [17,27,28].

Glasswort (*Salicornia herbacea* L.) is a halophytic plant that inhabits the mudflats of Korea and has been used as a seasoning and in folk medicine for intestinal ailments, nephropathy, and hepatitis [29]. The extract of glasswort prevents high fat diet-induced hyperglycemia and hyperlipidemia in mice and induces antioxidant and skin-whitening effects [30,31]. Methyl 3,5-dicaffeoyl quinate (MDQ), an active compound present in *S. herbacea.* L produces anti-melanogenic effects through p-p38 and p-ERK1/2 signaling in B16F10 mouse melanoma cells [32]. (9Z,11E)-13-Oxooctadeca-9,11-dienoic acid [13-KODE (13-oxo-ode)] is a compound present in tomato fruit; it has health benefits and acts as a peroxisome proliferator-activated receptor-α (PPARα) agonist [33]. Furthermore, linoleic acid (LA) is oxidized into 13-hydroperoxy-9Z, 11E-octadecadienoic acid (13-HpODE) by lipoxygenase, and 13-HpODE is reduced to 13-hydroyoctadecadienoic acid (13-HODE) by glutathione peroxidase. 13-KODE (13-oxo-ode) is derived from 13-HODE by hydroxy fatty acid dehydrogenase [34]. However, the anti-inflammatory effects of 13-KODE derived from *S. herbacea* L. in murine macrophages have not been studied yet.

Here, we examined the anti-inflammatory activity of 13-KODE derived from *S. herbacea* L. using murine macrophages. We isolated the antioxidant component using an antioxidant assay-based purification protocol and the isolated compound, (9Z,11E)-13-Oxooctadeca-9,11-dienoic acid (13-KODE) was found to exhibit anti-inflammatory activity. We showed that 13-KODE functions as an anti-inflammatory agent by modulating NF-κB, ROS, and Nfe2I2 signaling in murine macrophages

## 2. Materials and Methods

### 2.1. Materials

Silica gel 60A and gel filtration resin (Sephadex LH-20) were obtained from Millipore-Sigma (Burlington, MA, USA). Reversed-phase high-performance liquid chromatography (HPLC)-UV analysis of antioxidants was carried out on a Shimadzu 20A series HPLC system (Kyoto, Japan) (Core-facility center, Jeju, Korea). LPS from *E. coli* was obtained from InvivoGen (San Diego, CA, USA). Cell proliferation was done using the EZ-cytox Cell Viability Assay Kit (Wellbio, Seoul, Korea). We used 13-KODE purified from *S. herbacea* L. extracts. All other chemicals were obtained from Millipore-Sigma (Burlington, MA, USA).

### 2.2. Plant Source

*S. herbacea L.* was purchased from Dasarang, Ltd. (Sinan, Korea). The plants were ground and lyophilized. A lyophilized sample (no. 2020_10) was deposited at the Department of Biomaterials, Jeju National University (Core-facility center, Jeju, Korea).

### 2.3. Purification of (9Z,11E)-13-Oxooctadeca-9,11-dienoic Acid (13-KODE)

The ground sample of *S. herbacea* L. (1000 g) was incubated with absolute methanol at 28 °C for 16 h. The antioxidant-based isolation is described in Figure 1A. We mixed methanol-extracted samples with distilled water and then evaporated the 100% methanol off using a rotary evaporator (Heidolph, Schwabach, Germany). The water-soluble portion was extracted with 1X ethyl acetate (EA). We isolated and then evaporated the EA fraction. The evaporated purified samples were dissolved in 100% methanol. The EA extracts were applied onto a silica gel resin (column size; 30 mm × 300 mm) and isolated with chloroform:methanol (30:1) (Appendix A). The extracts were divided into several fractions, evaporated, dissolved in methanol, and tested for antioxidant activity. The #6 fraction showed an antioxidant effect; it was applied to a Sephadex LH-20 gel (column size; 30 mm × 300 mm) and separated into four parts (Appendix A). Each part was evaluated for antioxidant activity and part #2 exhibited an effect. Part #2 was then loaded onto a preparatory thin-layer chromatography (TLC) plate (glass plate; 200 mm × 200 mm). The separated bands were examined for antioxidant activity (Appendix A). Fraction #1 was injected onto an HPLC column (Shimadzu LC-20A, Kyoto, Japan) (Shim-pack GIS C-18 column and elution rate; 2 mL/min) and a gradient elution was employed as follows: 0–60% acetonitrile for 20 min, 60–100% acetonitrile 10 min, and 100% acetonitrile for 20 min. The injection volume was 500 µl (Appendix A). The purified peak was observed at 32.8 min (Figure 1B).

### 2.4. Structural Analysis of the Isolated Sample

The chemical structure of the isolated compound was determined by mass spectrometry and NMR. The molecular weight was established as 294 by ESI-mass spectrometry, which showed quasi-molecular ion peaks at *m*/*z* 295.4 [M+H]+ in positive mode and *m*/*z* 293.4 [M-H]- in negative mode (see Appendix A). The 1H NMR spectrum in CDCl3 exhibited signals resulting from 4 olefinic methines at δ 7.48, 6.17, 6.11, and 5.88, 11 methylenes at δ 2.53, 2.34, 2.29, 1.62, 1.62, 1.41, and 1.25–1.35, and 1 methyl at δ 0.88. In the 13C NMR spectrum, 18 carbon peaks were observed, including 2 carbonyl carbons at δ 201.4 and 178.3, 4 olefinic methine carbons at δ 142.6, 137.0, 129.2, and 127.0, 11 methylene carbons at δ 41.3, 33.8, 31.5, 28.7–29.1, 28.2, 24.6, 24.1, and 22.5, and 1 methyl carbon at δ 13.9 (see Appendix A). All proton-bearing carbons were assigned by the Heteronuclear Multiple Quantum Coherence (HMQC) spectrum, and the 1H-1H Correlated Spectroscopy (COSY) spectrum revealed four partial structures, CH3-CH2-, -CH2-CH2-, -CH=CH-CH=CH-CH2-CH2-, and -CH2-CH2-CH2- (see Appendix A). Further structural elucidation was performed using the Heteronuclear Multiple Bond Correlation (HMBC) spectrum, which showed long-range correlations from the methyl protons at δ 0.88 to the carbons at δ 31.5 and 22.5, from the methylene protons at δ 2.53 to the carbons at δ 31.5 and 24.1, and from the protons at δ 6.17 and 2.53 to the ketone carbonyl carbon at δ 201.4. The methylene protons at δ 2.34 and 1.62 showed a long-range correlation to the carbon at δ 178.3 (see Appendix A). Finally, the structure of the isolated compound was determined as (9Z,11E)-13-Oxooctadeca-9,11-dienoic acid (13-KODE) by the process of elimination. The geometries of C-9 and C-11 were established as cis and trans, respectively, by the proton coupling constants of 11.0 and 15.5 Hz.

### 2.5. Antioxidant Assay

The antioxidant assay was done by the 2,2-Diphenyl-1-picrylhydrazyl (DPPH) method [35]. Several concentrations of 13-KODE (0, 25, 50, 75, and 100 µM) and a representative antioxidant, 10 mM N-acetyl-cysteine (NAC) as a positive control, were incubated with 200 µM DPPH solution in a 96-well plate for 20 min. The absorbance at OD_517_ was examined using a VersaMax plate reader (Molecular Devices, San Jose, CA, USA). The DPPH scavenging activities were determined by the following equation: DPPH scavenging activity (%) = {1-[(Sample-Blank)/Control]} × 100.

### 2.6. Cell Line and Culture Conditions

RAW 264.7 macrophages were purchased from the American Type Culture Collection (Manassas, VA, USA). The macrophage cells were cultured in RPMI-1640 medium with 10% Cytiva HyClone fetal bovine serum (Marlborough, MA, USA) and 1% Penicillin/Streptomycin (Cytiva, HyClone) at 37 °C in 5% CO_2_.

### 2.7. Cell Proliferation Assay

RAW 264.7 cells (2.5 × 10^6^ cells/plate) were seeded into a 96-well plate for 24 h. The murine macrophages were incubated with increasing concentrations of 13-KODE (0, 25, 50, 100, 200, 300, and 400 µM) for one day. Cell proliferation was determined using the EZ-Cytox Cell Viability Assay Kit (Wellbio, Seoul, Korea) following the manufacturer’s instructions. The cultured media and reagent solution were mixed at a 10:1 ratio and 100 µL of the mixture was added to each well and incubated at 37 °C for 2 h. The absorbance at OD_492_ was measured using a VersaMax plate reader (Molecular Devices).

### 2.8. NO Assay

RAW 264.7 macrophages (2.5 × 10^6^ cells/plate) were incubated in a 12-well plate for 24 h. The macrophages were incubated with two concentrations of 13-KODE (0 or 100 µM) without LPS or with several concentrations of 13-KODE (0, 25, 50, 75, and 100 µM) with 1 µg/mL LPS for 1 day. Secretory NO concentrations were measured using the NO Plus Detection kit (LiliF, Gyeonggi, Korea). Then, 100 µL of the media and nitrite standard were incubated by adding 50 µL of N1 buffer to each reaction for 30 min, and the combined solution was incubated with 50 µL of N2 buffer for 15 min. The production of NO was determined by measuring the absorbance OD_560_ using a VersaMax microplate reader (Molecular Devices).

### 2.9. Quantitative Real Time Reverse Transcription Polymerase Chain Reaction (qRT-PCR)

RAW 264.7 macrophage cells were seeded into a 6-well plate for 24 h. The cells were treated with 13-KODE (50 and 100 µM) for 1 h and incubated with LPS (1 µg/mL) for 24 h. Total RNA was extracted using the RNAiso Plus Extraction Kit (TaKaRa, Tokyo, Japan) following manufacturer’s instructions. Real time qRT-PCR was carried out using a One Step PrimeScript RT-PCR kit (TaKaRa, Tokyo, Japan). The qRT-PCR reaction mixture contained 10 µL of 2X One Step RT-PCR buffer, 0.5 µL of PrimeScript RT enzyme Mix, 0.5 µL of Takara ExTaq, 1 µL of total RNA (150 ng/µL), 1 µL of forward primer (10 ng/µL), 1 µL of reverse primer (10 ng/µL), and 6 µL of RNase-free sterile water. We used the comparative C_t_ method for analyzing the relative transcript levels of the target genes. The specific qRT-PCR primers were obtained from Bioneer (Daejeon, Korea) and described in Appendix A. The β-actin gene was used as an internal control.

### 2.10. Enzyme-Linked Immunosorbent Assay (ELISA) for Cytokines

RAW 264.7 macrophage cells were seeded in a 12-well plate. The cells were pretreated with 13-KODE (50 and 100 µM) for 1 h and stimulated with LPS (1 µg/mL) for 24 h. The amount of TNF-α and IL-1β in the supernatant medium was measured using a specific ELISA kit. The amount of IL-1β and TNF-α was measured using the IL-1β Mouse and TNF-α ELISA Kit (R&D, Minneapolis, MN, USA).

### 2.11. Western Blot Analysis

RAW 264.7 macrophage cells were treated with 13-KODE for 1 h and incubated with 1 µg/mL LPS for 30 min. The cells were lysed using a radioimmunoprecipitation assay (RIPA) buffer (Sigma-Aldrich, Burlington, MA, USA) supplemented with protease inhibitor, 10 mM sodium fluoride, and 10 mM sodium vanadate. Each lysate was separated using sodium dodecyl sulfate polyacrylamide gel electrophoresis (SDS-PAGE), and the proteins were transferred to polyvinylidene fluoride (PVDF) membranes (Sigma-Aldrich, Burlington, MA, USA). After incubating with Odyssey^®^ Blocking Buffer (PBS) (LI-COR, Lincoln, NE, USA) at room temperature for 1 h, the membranes were incubated for 3 h at room temperature with primary antibodies. After washing three times with 1X PBS containing 0.1% Tween 20, the membrane was incubated with IRDye^®^ 680RD- and IRDye^®^ 800CW-labeled antibody in Odyssey blocking buffer (1X PBS) supplemented with 1X PBS containing 0.1% Tween 20 at room temperature for 1 h. The protein bands were visualized using Odyssey CLx (LI-COR, Lincoln, NE, USA). Anti-pp38, anti-JNK, anti-pJNK, anti-ERK1/2, anti-pERK1/2, anti-p65, anti-NRF2 (Nfe2l2), and anti-HO-1 antibodies were obtained from Cell Signaling Technology (Beverly, MA, USA). Anti-p38, anti-β-actin, and anti-Lamin B antibodies were obtained from Santa Cruz Biotechnology (Dallas, TX, USA).

### 2.12. Immunofluorescence Staining

RAW 264.7 macrophage cells (5 × 10^5^ cells/mL) were incubated in a 96-well black plate with a glass bottom (Eppendorf, Hamburg, Germany). The cells were pretreated with 13-KODE for 1 h and stimulated with LPS (1 µg/mL) for 60 min. The cells were fixed in 4% paraformaldehyde for 15 min, permeabilized with 0.1% Triton X-100 for 5 min, and treated with 50 mM NH_4_Cl for 5 min. The cells were blocked with PBS containing 3% BSA for 30 min and stained with p65 antibody (Cell Signaling Technology) overnight at 4 °C. The stained cells were washed with 1X PBS and incubated with goat anti-mouse IgG Alexa 488-conjugated secondary antibodies. The cells were subsequently stained using mounting medium with DAPI (Abcam, Cambridge, UK) and detected with an automated microscope (Lionheart, Biotek, VT, USA).

### 2.13. Determination of Cellular ROS by Invitrogen^®^ CellROX^®^ Green Reagent

The cellular ROS concentration was determined using Invitrogen CellROX^®^ Green Reagent (Invitrogen, Carlsbad, CA, USA) following the manufacturer’s instructions. RAW 264.7 macrophage cells (1 × 10^6^ cells/plate) were seeded in a 96-well plate for 24 h. The cells were pretreated with 13-KODE or N-acetyl-L-cysteine (NAC) for 60 min and incubated with LPS (1 µg/mL) for 30 min. Then, the cells were stained with CellROX green dye for 10 min at 37 °C. After washing with 1X PBS, the stained ROS were captured using an automated microscope (Lionheart, Biotek, VT, USA).

### 2.14. Statistical Analysis

The data was the result of three independent experiments and reported as the mean ± standard deviation. All data were analyzed using one-way analysis of variance and evaluated using GraphPad Prism 8.0 (GraphPad Software Inc., San Diego, CA, USA), and ** *p* < 0.01, *** *p* < 0.001 was considered a significant statistical difference.

## 3. Results

### 3.1. Purification of an Antioxidant Derived from S. herbacea L.

The antioxidant-guided (DPPH assay) purification steps are presented in Figure 1A. The ground powder of *S. herbacea* L. was extracted with 100% methanol, and the dried samples of *S*. *herbacea* L. were extracted with EA and H_2_O (*v/v* = 1:1). The EA extracts were isolated using silica gel chromatography, Sephadex LH-20 gel chromatography, preparatory thin layer chromatography (prep-TLC), and HPLC (Figure 1A and Appendix A). The isolated sample was shown using HPLC and it has antioxidant activity (Figure 1B).

### 3.2. Structure of Isolated Antioxidant and the Effect of 13-KODE on Cell Proliferation

The purified antioxidant was identified as 13-Oxo-9Z, 11E-octadecadienoic acid (13-KODE) using NMR spectrometry (Figure 1C and Appendix A). Cell proliferation was assessed using 13-KODE in murine RAW 264.7 macrophages. The cells were incubated with 13-KODE up to 400 μM. The results indicated that 13-KODE (100 μM) without LPS, LPS (up to 10 μg) without 13-KODE, and the combination of LPS and 13-KODE exhibited no growth inhibition on macrophage cells (Figure 2A,B), whereas 200 μM 13-KODE inhibited cell proliferation by 42%.

### 3.3. Effect of 13-KODE on LPS-Stimulated Production of Proinflammatory Mediators in RAW 264.7 Cells

To determine the suppressive effect of 13-KODE on LPS-stimulated NO secretion, we tested murine macrophage cells with 13-KODE (25, 50, 75, and 100 μM) for 1 day with or without LPS. It was observed that 13-KODE (100 μM) without LPS did not change NO secretion of macrophages (Figure 2D). Additionally, we found that LPS-induced NO secretion and 13-KODE decreased LPS-stimulated NO secretion. LPS stimulation increased NO secretion up to 42-fold, whereas 13-KODE treatment resulted in a 21%, 49%, 70%, and 90% reduction, respectively, compared with LPS-stimulated cells (Figure 2D). We examined the expression of iNOS using an immunoblot assay. The LPS-treated RAW 264.7 macrophages increased the expression of iNOS, whereas 13-KODE markedly reduced LPS-induced iNOS protein levels. The mRNA level of iNOS was measured in LPS-treated RAW 264.7 macrophages. LPS-induced mRNA levels of the iNOS gene, whereas 13-KODE treatment resulted in a decrease in iNOS mRNA (Figure 2E). Thus, 13-KODE decreased NO secretion by decreasing iNOS gene expression.

### 3.4. 13-KODE Inhibits LPS-Stimulated Proinflammatory Cytokines in RAW 264.7 Macrophage Cells

The effect of 13-KODE on the elevated levels of TNF-α and IL-1β mRNA and protein in RAW 264.7 macrophage cells was determined by qPCR and ELISA. LPS induced mRNA and protein levels of IL-1β by 88- and 7.4-fold, respectively, whereas 13-KODE (100 μM) down-regulated expression by 52% and 72% compared with the LPS-treated control (Figure 3A). LPS induced transcripts and protein levels of TNF-α by 6.3- and 9.8-fold, respectively, whereas 13-KODE (100 μM) treatment resulted in a 66% and 61% decrease compared with LPS-stimulated cells (Figure 3B). We found that 13-KODE decreased IL-1β and TNF-α secretion by inhibition of IL-1β and TNF-α gene expression (Figure 3A,B). LPS stimulation markedly induced the secretion of TNF-α and IL-1β in RAW 264.7 macrophages, whereas 13-KODE decreased LPS-induced production of TNF-α and IL-1β (Figure 3A,B). LPS induced the expression of TNF-α and IL-1β genes, whereas 13-KODE reduced LPS-stimulated TNF-α and IL-1β gene expression (Figure 3A,B). These results indicate that 13-KODE decreases the inflammatory response of RAW 264.7 macrophage cells by reducing LPS-induced cytokine secretion.

### 3.5. The Effect of 13-KODE on NF-κB Signaling

We determined the effects of 13-KODE on the NF-κB signaling pathway. The nuclear translocation of NF-κB p65 in LPS-treated cells was increased 1.6-fold compared with control cells, whereas 13-KODE reduced the levels of nuclear p65 by 67% compared with LPS-stimulated cells. The expression of cytosolic IκB was induced by 2.6-fold compared with the LPS-stimulated control. We found that LPS stimulation increased the nuclear level of NF-κB p65 protein and down-regulated the expression of IκB-α. Moreover, 13-KODE treatment reduced the nuclear level of NF-κB (p65) and inhibited the downregulation of IκB-α expression in LPS-stimulated cells (Figure 4A). The immunofluorescence data revealed that LPS-treatment induced the nuclear translocation of NF-κB p65, whereas pretreatment with 13-KODE inhibited the nuclear translocation of NF-κB p65 in LPS-treated cells (Figure 4B). We demonstrated that 13-KODE is a suppressor of LPS-stimulated NF-κB activation in RAW 264.7 macrophage cells.

### 3.6. 13-KODE Inhibits LPS-Induced MAPK Activation in RAW 264.7 Macrophages

Our data showed that LPS stimulation increased the levels of pERK1/2, p-p38, and pJNK by 4.8-, 5.5- and 7-fold, respectively, whereas 13-KODE resulted in a decrease of 66%, 47%, and 52%, respectively, compared with LPS-stimulated control cells (Figure 5). We found that 13-KODE suppressed LPS-treated levels of pERK, p-p38, and pJNK (Figure 5) and inhibited the inflammatory response of LPS-induced RAW 264.7 cells by inhibiting MAPK signaling.

### 3.7. 13-KODE Reduced LPS-Induced ROS Accumulation in RAW 264.7 Cells

Figure 6 shows that the LPS-induced cellular signal of CellROX Green was reversed by 13-KODE and NAC. Fluorescent signals were increased by 12.2-fold in LPS-treated cells and reduced by 88% in 13-KODE–treated cells. This suggests that 13-KODE is a strong candidate for ROS-reducing activity against LPS on murine RAW 264.7 macrophages.

### 3.8. Induction of HO-1 and Nfe2I2 Expression by 13-KODE in RAW 264.7 Cells 

The Nrf-2 (Nfe2I2) and HO-1 signaling axis represents a multiorgan protector that decreases oxidative stress in tissue and animal models [17,28]. We determined whether 13-KODE could induce the Nfe2I2/HO-1 signaling axis. 13-KODE increased HO-1 and Nfe2I2 protein levels by 20- and 4.6-fold, respectively, and reduced the expression of Keap1 protein at 12 h by 67%. We found that 13-KODE induced HO-1 and Nfe2I2 protein levels (Figure 7A). We determined the cytosolic/nuclear location of Nfe2I2 during 13-KODE treatment and found that nuclear Nfe2I2 protein was increased by 5.3-fold. 13-KODE significantly increased the level of nuclear Nfe2I2 protein in nuclear fractions of RAW 264.7 macrophage lysates (Figure 7B). The results indicate that the antioxidant effect of 13-KODE is associated with the Nfe2I2/HO-1 signaling axis.

## 4. Discussion

*S. herbacea* L. (glasswort) inhabits the western coastline of Korea; it is utilized as a food resource and consumed as a raw vegetable in salads and fermented food [36,37]. *S. herbacea* L. exhibits several biological effects including antioxidant, anti-inflammatory, and anticancer activity [37,38,39,40]. In the present study, we isolated an antioxidant from an *S. herbacea* L. extract using an antioxidant assay purification method, and the antioxidant was identified as 13-KODE, a 13-Oxo-9(Z), 11(E)-octadecadienoic acid. We demonstrated that 13-KODE extracted from glasswort exhibits anti-inflammatory activity on LPS-stimulated RAW 264.7 macrophages. 13-KODE (13-oxo-ode) is derived from 13-HODE by hydroxy fatty acid dehydrogenase [34]. LA is an essential fatty acid, and its derivatives consist of 9,10-epoxy-12-octadecenoic acid (leukotoxin), 12,13-epoxy (EKODE), 9-hydroxyoctadacadienoic acid (9-HODE), 13-hydroxyoctadacadienoic acid (13-HODE), and 13-oxo-octadecadienoic acid (13-oxo-ode, 13-KODE). The derivatives of LA are synthesized by cyclooxygenases, lipoxygenases, and cytochrome p450 and are known to have pleiotropic effects [41].

The anti-inflammation properties of 13-KODE derived from *S. herbacea* L. have not been studied in murine macrophages. We examined the protective effect of 13-KODE on LPS-stimulated RAW 264.7 macrophages. LPS extracted from the cell wall of Gram negative bacteria may be used as a tool to determine the ability of specific compounds to inhibit inflammation [4]. After treatment with LPS, murine macrophages secrete inflammatory cytokines, such as TNF-α, IL-6, and IL-1β. In the present work, 13-KODE inhibited LPS-stimulated NO production, TNF-α secretion, and IL-1β secretion. It suppressed LPS-stimulated inflammation. The NF-κB pathway is essential for producing an inflammatory response and increases the expression of several proinflammatory genes [42]. This pathway is a target for the treatment of inflammatory diseases [43]. Our results indicated that 13-KODE inhibited the LPS-stimulated nuclear translocation of NF-κB (p65). We also found that 13-KODE suppressed LPS-stimulated MAPK activation (pERK1/2, pJNK, and p-p38) and LPS-stimulated inflammation by modulating MAPK signaling and NF-κB (p65) nuclear translocation.

In murine macrophage cells, LPS-induced cellular ROS accumulation increased the inflammatory mediators, and cytokine secretion [44]. The crosstalk between inflammation and oxidative stress is important in diseases. LPS-induced oxidative stress and inflammation in bovine mammary epithelial cells and acute lung injury were inhibited by hydroxytyrosol and adamantly retinoid ST1926 [19,20,21]. Our data showed that LPS-induced oxidative stress and inflammation in RAW 264.7 macrophage cells were inhibited by 13-KODE. It is considered that hydroxytyrosol, retinoid ST1926, and 13-KODE compounds can regulate the crosstalk between inflammation and oxidative stress. ROS acts as a mediator of receptor-mediated signaling. Lipoxygenases (LOXs) and nicotinamide adenine dinucleotide phosphate oxidases of immune cells induce the production of ROS during receptor-mediated signaling. LPS-treatment increases ROS production and NF-κB activation [44]. We demonstrated that 13-KODE reduced LPS-stimulated ROS production in murine macrophages and inhibited ROS accumulation, which represents an important anti-inflammatory effect of 13-KODE.

The Nfe2I2/HO-1 axis is a multiorgan protective mechanism that protects cells against oxidative stress. It inhibits the transcriptional activation of proinflammatory cytokines and inflammatory genes [45] as well as oxidative stress in cells and organs [46]. The Nfe2I2/sMaf transcription factor complex binds to the antioxidant response element and activates the transcription of the HO-1 gene. The bilirubin produced by HO-1 exhibits antioxidant and cytoprotective effects. We showed that 13-KODE increased Nfe2I2 and HO-1 protein levels and Nfe2I2/HO-1 axis signaling. It also increased the anti-inflammatory response of LPS-induced macrophage cells. Furthermore, we also found that 13-KODE increased Nfe2I2 and HO-1 protein levels and decreased Keap1.

13-KODE, a derivative of LA, has been isolated from various tomato plants. It exhibits antioxidant effects and acts as a PPARα agonist [33,34]. 13-HODE, a precursor of 13-KODE act as a mitogenic signal responsible for LA-dependent growth in hepatoma 7288CTC cells in vivo [47]. 13-KODE shows structural stability under hot and acidic conditions. Our results showed that 13-KODE exhibits anti-inflammatory effects on murine RAW 264.7 cells not only by blocking ROS accumulation, MAPK activities and activating NF-κB signaling, but also by increasing Nrf-2/HO-1 signaling (Figure 8). Our study showed that 13-KODE results in an anti-inflammatory effect and may have the potential for treating inflammatory diseases.

## 5. Conclusions

We purified an antioxidant compound from *S. herbacea* L. (glasswort) and found that it exhibits significant antioxidant activity. Purification of the antioxidant compound was carried out through antioxidant assay-based isolation and large sample preparation. Using NMR spectrometry, we identified the purified compound to be 13-KODE. We determined that it has a protective effect on LPS-stimulated inflammation of macrophages. 13-KODE exhibits its anti-inflammatory activity in LPS-stimulated macrophages by inhibiting proinflammatory cytokine production, LPS-stimulated ROS accumulation, LPS-induced MAPK activation and activating Nfe2I2/HO-1 signaling axis. We showed that 13-KODE produces anti-inflammatory effects and has the potential for use in treating inflammatory diseases.

## Figures and Tables

**Figure 1 antioxidants-11-00180-f001:**
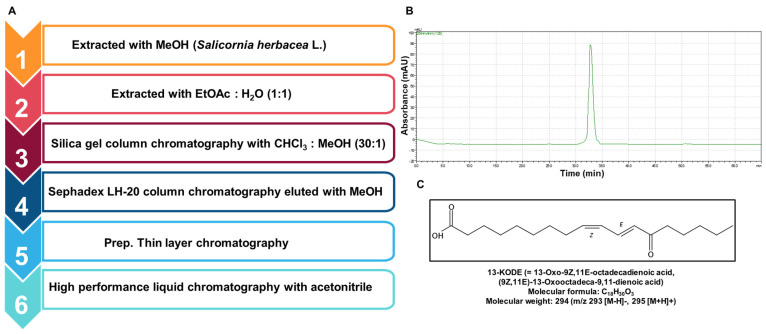
Procedure for the isolation of an antioxidant from *S. herbacea* L. and the molecular structure of 13-KODE. (**A**) Purification flowchart of the antioxidant from *S. herbacea* L. (**B**) HPLC analysis of an antioxidant derived from *S. herbacea* L. (**C**) Chemical structure of 13-KODE, the antioxidant purified from *S. herbacea* L.

**Figure 2 antioxidants-11-00180-f002:**
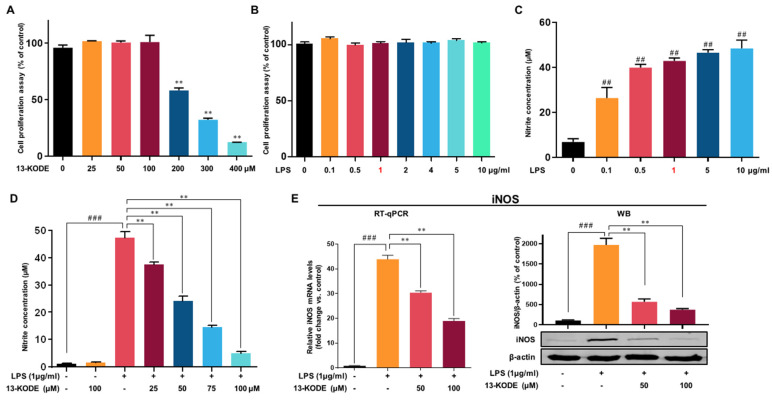
Effect of 13-KODE on cell proliferation and LPS-induced NO secretion and expression of the iNOS gene in RAW 264.7 macrophages. The cells were cultured with the indicated concentrations of 13-KODE and LPS. (**A**,**B**) Cell proliferation was determined with WST (4-[3-(4-Iodophenyl)-2-(4-nitro-phenyl)-2H-5-tetrazolio]-1,3-benzene sulfonate). The results are shown as the percentage of surviving cells compared with control cells (without 13-KODE and LPS). (**C**) RAW 264.7 macrophage cells were treated with several concentrations of LPS for 24 h, and NO production was determined using the NO assay. (**D**) RAW 264.7 cells (4 × 10^5^ cells/well in six-well plates) were left untreated or were pretreated with 13-KODE (25, 50, 75, and 100 μM) prior to treatment with LPS (1 μg/mL) for 24 h. (**E**) Total RNA was purified from RAW 264.7 macrophage cells with or without the indicated concentrations of 13-KODE and treated with LPS (1 μg/mL) for 24 h. The transcripts of iNOS were measured by quantitative reverse transcription polymerase chain reaction as described in Section 2. Protein lysates were isolated from the cells with or without 13-KODE and then stimulated with LPS (1 μg/mL) for 24 h. The values correspond to the mean ± standard deviation, *n* = 3. ^##^
*p* < 0.01; ^###^
*p* < 0.001; ** *p* < 0.01.

**Figure 3 antioxidants-11-00180-f003:**
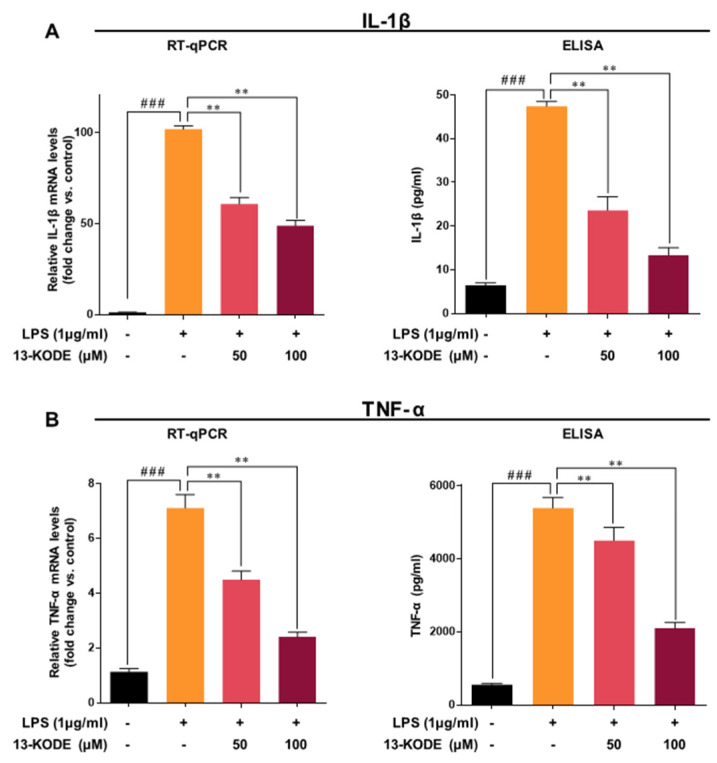
Inhibition of IL-1β and TNF-α production by 13-KODE in LPS-stimulated RAW 267.4 cells. RAW 264.7 cells were incubated with 13-KODE (50 and 100 μM) prior to stimulation with 1 μg/mL LPS for 1 day and total RNA was isolated. The transcript levels of IL-1β (**A**) and TNF-α (**B**) were measured by qRT-PCR. RAW 264.7 cells were incubated with 13-KODE (50 and 100 μM) prior to stimulation with 1 μg/mL LPS for 1 day and of IL-1β (**A**) and TNF-α (**B**) amounts in the cultured supernatants were measured using commercial ELISA kits. The values correspond to the mean ± standard deviation, *n* = 3. ** *p* < 0.01; ^###^
*p* < 0.001.

**Figure 4 antioxidants-11-00180-f004:**
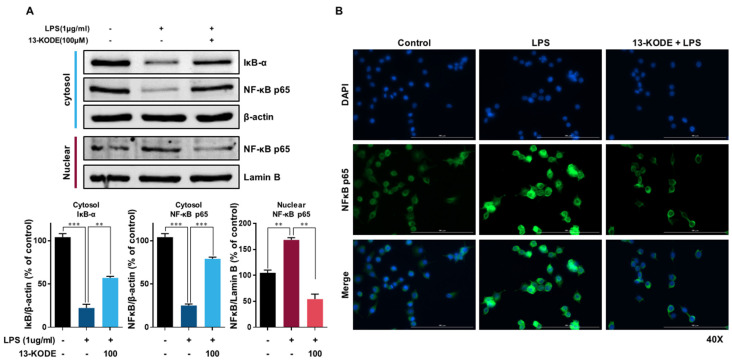
13-KODE inhibits LPS-induced nuclear translocation of NF-κB in RAW264.7 cells. Macrophage cells were treated with 100 μM 13-KODE for 1 h followed by treatment with LPS (1 μg/mL) for 30 min. (**A**) Nuclear and cytosolic fractions were subjected to SDS-PAGE followed by Western blotting using the indicated primary antibodies. The amounts of lamin B and β-actin were used as internal controls for the nuclear and cytosolic fractions, respectively. The values correspond to the means ± standard deviations, *n* = 3. ** *p* < 0.01 *** *p* < 0.001. (**B**) RAW264.7 cells were pretreated with 100 µM 13-KODE for 1 h and then treated with LPS (1 μg/mL) for 30 min. The localization of NF-κB p65 and nuclei were determined by staining with anti-p65 (green) and DAPI (blue). Images were obtained by microscopy at 40X magnification and show representative macrophages (scale bar = 100 μm).

**Figure 5 antioxidants-11-00180-f005:**
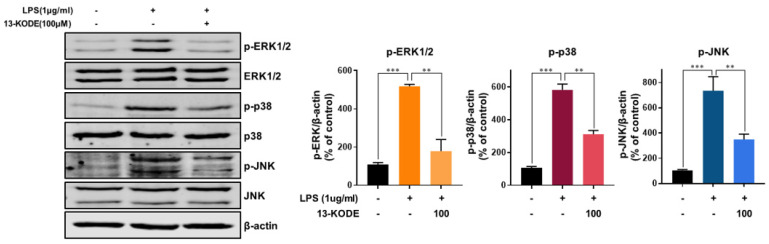
Inhibition of the LPS-induced activation of MAPKs by 13-KODE in RAW 264.7 macrophages. The cells were treated with 100 μM of 13-KODE for 1 h prior to exposure to LPS (1 μg/mL) for 30 min and total protein was isolated. The protein extracts were subject to SDS-PAGE followed by Western blot analysis using the indicated primary antibodies. The amount of β-actin served as the internal control. pERK1/2, p-p38, and pJNK levels were determined. The values represent the mean ± standard deviation, *n* = 3. ** *p* < 0.01; *** *p* < 0.001.

**Figure 6 antioxidants-11-00180-f006:**
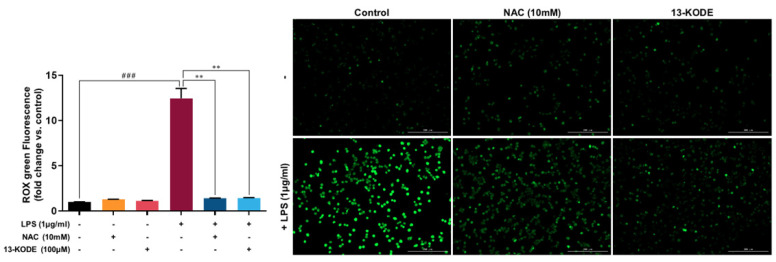
Effect of 13-KODE on LPS-induced ROS generation in RAW 264.7 macrophage cells. The ROS production of RAW 264.7 macrophages was monitored using CellROX Green dye assays. Macrophage cells were pretreated with 100 μM of 13-KODE and NAC (10 mM) for 1 h prior to treatment with or without 1 μg/mL LPS for 30 min. ROS data were visualized using a fluorescence microscope (magnification, ×100) and representative photos are shown (scale bar; 100 μm). The values represent the mean ± standard deviation, *n* = 3. ** *p* < 0.01 ^###^
*p* < 0.001.

**Figure 7 antioxidants-11-00180-f007:**
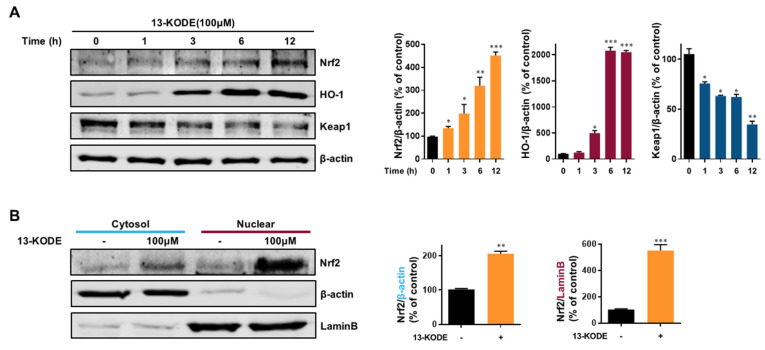
Induction of Nrf2 (Nfe2I2) and HO-1 proteins by 13-KODE in RAW 264.7 cells. (**A**) Macrophage cells were cultured with 100 μM 13-KODE for the indicated times. Total protein was isolated by 10% SDS-PAGE and transferred to PVDF membranes. The membranes were probed with the indicated antibodies. β-Actin served as the internal control. (**B**) Macrophage cells were treated with 100 μM 13-KODE for 12 h. Nuclear and cytosolic proteins were probed with anti-Nfe2I2 (Nrf2) antibody. Proteins were visualized using the near-infrared (NIR) Western blot system. Lamin B and β-actin were used as internal controls for the nuclear and cytosolic proteins. The values represent the mean ± standard deviation, *n* = 3. * *p* < 0.05; ** *p* < 0.01; *** *p* < 0.001.

**Figure 8 antioxidants-11-00180-f008:**
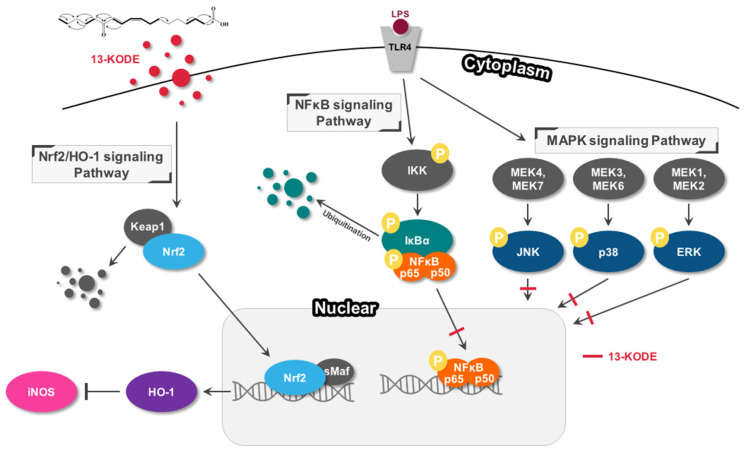
Scheme showing the mechanisms of the anti-inflammatory and cytoprotective effects of 13-KODE on LPS-induced inflammation.

## Data Availability

Data is contained within the article and Appendix A.

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
