# Peer review of "Anti-Inflammatory Effects of (9Z,11E)-13-Oxooctadeca-9,11-dienoic Acid (13-KODE) Derived from *Salicornia herbacea* L. on Lipopolysaccharide-Stimulated Murine Macrophage via NF-kB and MAPK Inhibition and Nrf2/HO-1 Signaling Activation"

_antioxidants, 2022, doi:10.3390/antiox11020180_

Round 1
Reviewer 1 Report
The authors investigated about the anti-inflammatory effects of 13-KODE on LPS-stimulated murine macrophages.
Overall the topic could be interesting but many details are not clear.
I recommend that the paper be accepted with minor revision:
a). The authors should mentioned in the abstract more details about model used.
b) In the introduction section, little previous evidence is provided about the importance of link between inflammation and oxidative stress. Incorporating comparisons with other studies would increase the strength of the paper. Please refer to doi: 10.3390/antiox9080693; 10.3892/ijmm.2018.3574; 10.3390/vetsci7040161.
c) The references list is not up to date. The authors should add more recent references.
d) There are some minor grammar issues that should be fixed in order to aid the accessibility of the results to the reader..
Author Response
We greatly appreciate your positive comments. Those comments are all valuable and very helpful for revising and improving our paper, as well as the important guiding significance to our researches. We have studied comments carefully and have made correction which we hope meet with approval. Revised portion are marked in blue in the paper.

Reviewer 2 Report
The manuscript by Ko et al. investigates the antioxidant and anti-inflammatory potential of 13-KODE, an antioxidant molecule the authors have extracted from Salicornia herbacea L. The authors studied the anti-inflammatory effects of 13-KODE using the murine macrophage cell line RAW-264.7 and they have shown that 13-KODE inhibited the LPS-induced nitric oxide (NO) production by downregulating iNOS, TNF-α and IL-1β expression. 13-KODE also inhibited the LPS-mediated nuclear localization of NF-κB and the activation of MAP Kinases. In addition, the authors have shown that 13-KODE reduced the LPS-induced production of ROS and increased the expression of Nfe2l2 and heme oxygenase 1.
Although the murine macrophage model (and in particular NO production) is controversial as a model of human inflammation implying to be cautious in terms of potential applications in human health, this study is well conducted and the results are significant and support the conclusions brought by the authors. However, the writing of the text needs to be considerably improved to avoid numerous redundancies or repetitions. Some major and minor problems were also identified as detailed below.
Major specific concerns:
- Line 113 and Figure 1C: Figure 1C is exactly the same as Figure S10. It would be better to show here a simplified molecular structure of 13-KODE (i.e. without H1-H1 COSY and HMBC information) and keep the detailed information for Figure S10. Otherwise, if Fig1C remain unchanged H1-H1 COSY and HMBC should be explained in the legend.
- Line 176, Table S1 is missing.
- Lines 237-238 and Figure 2A,B: it is not clear that cell proliferation is assessed with LPS alone and 13-KODE alone. Please rephrase. In addition, please add the result of cell proliferation assay with the combination of LPS and 13-KODE (as for NO production).
- Lines 31 and 419 are overinterpretation of the results. Please modulate this conclusion by using "may have" or "probably could have" since demonstration of therapeutic potential requires in-vivo experiments, which are absent from the present work.
Minor concerns:
1. After it first occurrence (i.e. line 58) please used the abbreviated spelling S. herbacea L. throughout the manuscript (i.e. lines 63, 70, 71, 85, 228, 229, 380 etc). Idem, in the abstract used the abbreviated name at line 24.
2. Lines 30 and 53: according to the most recent nomenclature please replace Nrf2 by Nfe2l2 (Nuclear factor erythroid-2 like 2) throughout the manuscript (i.e. lines 56, 76, 197, etc).
3. Line 30, please replace “/” by “and”
4. Lines 124-127, please define COSY, HMQC and HMBC
5. Line 136, please define DPPH
6. Line 216, please define NAC
7. Line 231, please develop the sentence.
8. Line 257, please define WST
9. Lines 268-269, these sentences are useless here (introduction and materials and methods).
10. Lines 291-293, these sentences are useless here (introduction and materials and methods).
11. Lines 296-298, these two sentences are particularly unclear and probably useless. Please rephrase or delete.
12. Line 315, the 3.6 heading should be positioned correctly (i.e. not in the legend of the Figure).
13. Lines 317-319, these sentences are useless (general information and materials and methods).
14. Line 333-335, this sentence is useless (already described in materials and methods section).
15. Line 368-374, please align the legend of the figure to the left.
16. Line 385, “in the present,” do you mean “in the present work”?
Author Response

(The authors gave the same response as above.)

Round 2
Reviewer 2 Report
The authors have made all the requested changes in a very satisfactory way and the reviewer thanks them very much for that. Nevertheless, Table S1 should be mentioned in the list of Supplementary Materials (line 446 and following). No other change required by this reviewer.